# Unique Features of the m^6^A Methylome and Its Response to Salt Stress in the Roots of Sugar Beet (*Beta vulgaris*)

**DOI:** 10.3390/ijms241411659

**Published:** 2023-07-19

**Authors:** Junliang Li, Qiuying Pang, Xiufeng Yan

**Affiliations:** 1National and Local Joint Engineering Research Center of Ecological Treatment Technology for Urban Water Pollution, Zhejiang Provincial Key Laboratory for Water Environment and Marine Biological Resources Protection, Institute for Eco-Environmental Research of Sanyang Wetland, College of Life and Environmental Science, Wenzhou University, Zhong-Xin Street, Wenzhou 325035, China; 20210417@wzu.edu.cn; 2Post-Doctoral Research Stations, Northeast Forestry University, Harbin 150040, China; 3Key Laboratory of Saline-Alkali Vegetation Ecology Restoration, Ministry of Education, College of Life Sciences, Northeast Forestry University, Harbin 150040, China

**Keywords:** m^6^A-sequencing, differentially methylated peaks, mRNA stability, salt stress, sugar beet

## Abstract

Salt is one of the most important environmental factors in crop growth and development. *N*^6^-methyladenosine (m^6^A) is an epigenetic modification that regulates plant–environment interaction at transcriptional and translational levels. Sugar beet is a salt-tolerant sugar-yielding crop, but how m^6^A modification affects its response to salt stress remains unknown. In this study, m^6^A-seq was used to explore the role of m^6^A modification in response to salt stress in sugar beet (*Beta vulgaris*). Transcriptome-wide m^6^A methylation profiles and physiological responses to high salinity were investigated in beet roots. After treatment with 300 mM NaCl, the activities of peroxidase and catalase, the root activity, and the contents of Na^+^, K^+^, and Ca^2+^ in the roots were significantly affected by salt stress. Compared with the control plants, 6904 differentially expressed genes (DEGs) and 566 differentially methylated peaks (DMPs) were identified. Association analysis revealed that 243 DEGs contained DMP, and 80% of these DEGs had expression patterns that were negatively correlated with the extent of m^6^A modification. Further analysis verified that m^6^A methylation may regulate the expression of some genes by controlling their mRNA stability. Functional analysis revealed that m^6^A modifications primarily affect the expression of genes involved in energy metabolism, transport, signal transduction, transcription factors, and cell wall organization. This study provides evidence that a post-transcriptional regulatory mechanism mediates gene expression during salt stress by affecting the stability of mRNA in the root.

## 1. Introduction

Soil salinization is a major problem in the maintenance of sustainable agriculture development [1,2]. According to the Status of the World’s Soil Resources (SWSR) released by the Food and Agriculture Organization (FAO) of the United Nations in 2015, more than 60 million hm^2^ of irrigated farmland in the world is affected by soil salinization, and 0.3 to 1.5 million hm^2^ of irrigated farmland loses its production capacity every year because of salinization [3]. Exploring salt-tolerant genes and breeding new varieties of salt-tolerant crops may be effective measures to address this problem.

Sugar beet (*Beta vulgaris*) is the second largest global contributor to sugar production, with about 30% of sugar being produced from it [4]. Under salt stress, beet can quickly synthesize and accumulate betaine, coordinate osmotic balance, and preserve normal physiological metabolism [5]. In addition, sugar beet can use Na^+^ to partially replace K^+^ in osmoregulation, support long-distance anion transport, control stomatal opening and closing, and regulate enzyme activities and chloroplast proliferation [6,7]. In general, plants can adjust to salt in three ways: by rejecting, secreting, and accumulating the salt. Sugar beet is a salt-accumulating type of plant, with a strong ability to accumulate salt. It can transport and store salt absorbed from the environment in its petioles and aged leaves, ensuring a lower salt content in the leaves with higher photosynthetic capacity [8]. In general, the salt tolerance of sugar beet does not depend on the particular structure of the halophytes, such as salt glands or succulent stems and leaves, but stems from the physiological and biochemical mechanisms and metabolic properties formed via adaptation to the environment during evolution. Consequently, the study of salt tolerance mechanisms in sugar beet will not only be beneficial for improving salt tolerance in the plant, but may also guide the improvement of other non-halo crops.

Although sugar beet shows strong tolerance to salt stress, the salt tolerance differs significantly for different varieties (genotypes). Skorupa et al. [9] investigated transcriptome changes under salt stress and found that stronger transcriptome changes are required to maintain homeostasis in sugar beet (*Beta vulgaris* cv. Huzar) than in sea beet (*Beta vulgaris* ssp. *maritima*) when both are subjected to high salinity during cultivation. These investigators also found that salt stress increases the expression of genes related to ribosome biogenesis, photosynthetic carbon fixation, and cell wall construction and expansion in sea beet [10]. A previous investigation showed that genes related to the regulation of redox balance, signal transduction, and protein phosphorylation were differentially expressed in the sugar beet monosomic addition line M14 when subjected to salt stress under different NaCl concentrations (200 and 400 mmol·L^−1^) [11]. Furthermore, changes in miRNAs, lncRNAs, and circRNAs were also observed in cultivated beet (*Beta vulgaris* cv. O68) under salt stress, and a ceRNA network has been constructed for these ncRNAs in response to salt stress [12]. Although a large number of salt-tolerant genes have been identified, how these salt-tolerant genes function under a unified regulatory network remains to be investigated. Recent studies have shown that epigenetic regulatory mechanisms may play an important role in regulating the expression of salt-tolerant genes, such as the modification of RNA by *N*^6^-methyladenosine.

*N*^6^-methyladenosine (m^6^A) is one of the most critical internal modifications of RNA [13,14]. It is a conserved and reversible post-transcriptional regulatory mechanism catalyzed by a methyltransferase (Writer), a demethylase (Eraser), and a reading protein (Reader) [15,16,17]. Recent studies have shown that m^6^A is found not only in mRNA but also in lncRNA and other non-coding RNAs, which have a large number of m^6^A modifications [18]. Studies in Arabidopsis have shown that m^6^A modification can enhance the stability of transcripts of specific abiotic stress response genes, ultimately increasing the protein levels of these stable transcripts [19]. The methylase gene (*virilizer*) mutation can markedly reduce the methylation level of salt-stress negative regulators (*ATAF1*, *GI*, *GSTU17*, etc.) and enhance the expression of these genes, and the mutant strains may exhibit a salt-sensitive phenotype [20]. While the methylation level of the *Atalkbh10B* mutant is significantly increased, a reduction in the expression of salt-stress negative regulators (*ATAF1*, *BGLU22*, *MYB73*, etc.), can delay the germination of the mutant seeds under salt stress, although the growth and survival rate of these mutants can be significantly improved [21]. Studies on maize and *Hippophae rhamnoides* have also found that drought stress can induce the expression of two and three *Atalkbh10B* homologs, leading to a decrease in the level of RNA methylation [22,23]. A growing number of studies have shown that epigenetic factors, especially m^6^A modification, play an influential role in the plant stress response. These findings suggest that m^6^A modification may be an essential epigenetic marker for regulating stress tolerance in plants. However, there have been few reports describing the m^6^A-mediated modulation of the salt stress response in sugar crops.

Sugar beets have been reported to have the ability to store salt absorbed from the environment in their cells to improve soil salinization [8]. Their roots are directly exposed to salt and play an important role in adaptation to salt stress. However, the dynamic response of m^6^A modifications to transcripts under salt stress in sugar beet roots is still unknown. In this study, m^6^A-seq was used to investigate changes in the RNA methylation profile of roots under salt stress in sugar beet. A total of 243 genes with different expression and methylation levels were identified. Of these, 80% showed a negative correlation between expression level and the extent of m^6^A modification, while 20% showed a positive correlation. These results reveal that m^6^A modification may extensively manage the salt stress response in sugar beet, primarily by negatively regulating the expression of salt tolerance genes.

## 2. Results

### 2.1. Physiological and Ionic Response to Salt Stress in Seedling Roots

Sugar beet seedlings were first treated with 300 mmol·L^−1^ NaCl for one day to investigate their response to salt stress, and various parameters of the plant were then determined to assess the extent of the response. Appendix A displays the phenotypic changes in seedling roots in the same pot before and after salt treatment. As for the activity of protective enzymes, plants treated with salt showed a seven-fold increase in peroxidase activity (Figure 1A and Appendix A), while displaying no significant changes in superoxide dismutase activity (Figure 1B), and a five-fold decrease in catalase activity (Figure 1C and Appendix A). In addition, a 1.5-fold increase in root activity (reduction capacity of TTC) was observed in salt-treated plants after one day of NaCl treatment, (Figure 1D and Appendix A). The proline content, which reflects the plant’s stress resistance, also increased moderately compared to the control plants (Figure 1E). As for the ionic response, the seedling roots showed a slight increase in sodium content after salt treatment (Figure 1F). Concurrently, the salt-treated plants displayed a significant decrease in the amount of potassium compared with the control plants (Figure 1G and Appendix A), but a significant increase in calcium content was observed in these plants (Figure 1H and Appendix A). In conclusion, salt treatment inhibited the uptake of potassium by the seedlings, induced proline accumulation, and enhanced POD activity and root activity.

### 2.2. Generation of m^6^A Methylation Profiles for Sugar Beet Roots

To identify the roles of m^6^A modification on salt tolerance in sugar beet, RNA sequencing without (input) and with m^6^A RNA immunoprecipitation (m^6^A-seq) was carried out, with three replicates for each sample. A total of 35 to 46 million reads were generated for each RNA-seq sample and 36 to 48 million reads were generated for each m^6^A-seq sample (Appendix A). The proportion of unique mapped reads for RNA-seq and m^6^A-seq was approximately 62 to 67%, with a Q30 greater than 93% for each sample. The average mapping rate of valid reads was 87.21%. These results reflect the tremendous depth and quality of the generated sequencing data. A total of 12,777 peaks of 11,233 genes were identified under the control condition (Appendix A), and 13,743 m^6^A peak callings of 11,923 genes were identified via salt stress (Appendix A), respectively. Among them, more than 86% of m^6^A-modified genes contained only one modified region, and less than 1% of m^6^A-modified genes contained more than two modified regions (Figure 2A). At the genome level, these m^6^A peaks were unevenly distributed across each chromosome (Figure 2B). To further characterize m^6^A in the transcripts of sugar beet, we investigated the metagene profile of the m^6^A peak. The result revealed that most m^6^A peaks were enriched near the stop codon and 3′ UTR (untranslated region) region of the genes (Figure 2C). Motif analysis indicated that more than 56% of RIP fragments contained the motif listed in Figure 2D, in accordance with canonical m^6^A modification sequence ‘URUm^6^AY’ (where R represents A/G and Y represents C/U) exclusive to plants [24]. Gene ontology (GO) analysis showed that m^6^A Peaks were greatly enriched among these genes in the following biological processes: ‘regulation of transcription, DNA-templated’, ‘transcription, DNA-templated’, ‘oxidation-reduction process’, ‘protein phosphorylation’ and ‘defense response’ (Figure 2E). Specifically, all these genes have protein binding activity (Figure 2E), which is necessary for their recognition and regulation by the m^6^A reader protein.

### 2.3. m^6^A Methylation Is Affected by Salt Stress

The methylation levels of the transcripts in the salt-treated and control plants were compared to explore the response of m^6^A modification to salt stress. A total of 6200 overlapping peaks were identified between the salt treatment and control conditions, of which 566 were defined as differentially methylated peaks (DMPs) according to |log2 FC| ≥ 1 and FDR < 0.05 (Figure 3A and Appendix A). Since the biological function of m^6^A modification depends on the modified gene itself, we analyzed the function of genes overlapping with DMPs via GO and KEGG analysis. GO terms including ‘defense response’, ‘transmembrane transport’, and ‘cell wall organization’ were particularly enriched in the genes overlapping with these DMPs (Figure 3B). KEGG analysis of the DMPs revealed that the differential m^6^A peak was mainly focused on genes related to ‘Cellular Processes’, ‘Environmental Information Processing’, ‘Genetic Information Processing’, ‘Metabolism’, and ‘Organismal Systems’ (Figure 3C). In detail, these five main classes were further subdivided into 18 sub-categories with 94 pathways, where the subclasses “Signal Transduction” and “Membrane Transduction” contain 2 (Plant hormone signal transduction and MAPK signaling pathway—plant) and 1 (ABC transporters) pathways, respectively (Figure 3C). The present results suggest that salt stress influences m^6^A modification levels in the root transcriptome, and that many genes with DMPs are associated with the salt stress response.

### 2.4. Conjoint Analysis of Changes in Gene Expression and Methylation Levels 

In order to investigate the effect of RNA methylation on gene expression under salt stress, the changes in gene expression and methylation levels were jointly analyzed. The gene expression levels (FPKM) were mostly comparable among the input samples (Figure 4A). A total of 6904 differentially expressed genes (DEGs, |log2 FC| ≥ 1 and *p* < 0.05) were identified by comparing the salt-treated plants with the control plants, with 2603 significantly up-regulated genes and 4301 significantly down-regulated genes (Figure 4B and Appendix A). Heatmap plots of these DEGs show the agreement of the three biological replicas and the significant differences between the salt-treated and control plant roots (Figure 4C). GO enrichment analysis found that DEGs were significantly enriched in several biological processes, some of which are related to salt stress, such as ‘defense response’, ‘cell surface receptor signaling pathway’, ‘hormone-mediated signaling pathway’, ‘cell wall biogenesis’, and ‘root cap development’ (Figure 4D). With respect to KEGG pathways, DEGs were significantly enriched in the signal transduction and basic cellular metabolism pathways, such as ‘Phenylpropanoid biosynthesis’, ‘Starch and sucrose metabolism’, ‘MAPK signaling pathway—plant’, and ‘Plant hormone signal transduction’ (Figure 4E).

Based on a combined analysis of the IP and input transcriptome sequencing, 243 DEGs were found to have DMPs by comparing the salt-treated with the control plants. In detail, 58 genes displayed m^6^A hypomethylation and up-regulated expression (hypo-up), while 24 genes had m^6^A hypomethylation and down-regulated expression (hypo-down), 29 genes had m^6^A hypermethylation and up-regulated expression (hyper-up), and 132 genes had m^6^A hypermethylation and down-regulated expression (hyper-down) (Figure 5A and Appendix A). These results indicate that the expression levels of more DEGs were negatively correlated (hypo-up and hyper-down: 190 genes) with the level of m^6^A modification under salt stress, while few DEGs were positively correlated (hyper-up and hypo-down: 53 genes). KEGG analysis revealed that genes whose expression levels were negatively correlated with m^6^A modification levels were primarily involved in Glycolysis/Glycogenesis (ko00010), fructose and mannose metabolism (ko00051), and the pentose phosphate pathway (ko00030) (Figure 5B), while genes whose expression levels were positively correlated with m^6^A modification levels were involved in gycosphingolipid biosynthesis (globo and isoglobo series (ko00603)), amino sugar and nucleotide sugar metabolism (ko00520), and protein processing in the endoplasmic reticulum (ko04141) (Figure 5C). These results suggest that m^6^A modification may be involved in the salt stress response by modulating carbohydrate metabolism.

### 2.5. Changes in RNA Methylation-Related Genes in Response to Salt Stress

To further explore the effect of m^6^A methylation on the response to salt stress in beet, we checked whether RNA methylation-modifying enzyme genes were also DEGs or DMPs. Based on our previous homology analysis [25], a total of 21 RNA methylation-related genes were collected in sugar beet, including five ‘m^6^A writers’, ten ‘m^6^A erasers’, and six ‘m^6^A readers’ (Appendix A). The results show almost no significant changes in the m^6^A modification levels of RNA methylation-related genes in sugar beet under salt stress, while only two RNA methylation-related genes were DEGs that responded salt stress (Table 1). *Bv6_150770_huzh* and *Bv7_179400_uxaj* encode two demethylases that are homologous to *AtALKBH1A* and *AtALKBH10A*, respectively. The down-regulation of these two genes may explain the reduced m^6^A levels in a portion of salt-tolerant genes.

### 2.6. m^6^A Modification Regulates mRNA Abundance by Regulating the Stability of Salt-Tolerant Transcripts

To investigate the effect of the extent of m^6^A modification on gene expression, time course experiments were performed to detect the stability of the target mRNAs. In the present study, nearly 80% of the 243 differentially expressed and methylated genes (Figure 5A and Appendix A) exhibited a negative correlation between expression level and the extent of methylation. Ten tolerance-related genes were selected from these genes to detect their mRNA retention rates. The results show that the mRNA retention rates of nine of these genes were significantly affected by the extent of m^6^A modification (Figure 6). Hypomethylation modification was found to increase the retention rates of fructose-bisphosphate aldolase 5 (*Bv4_091040_cyuu*), electron transfer flavoprotein-ubiquinone oxidoreductase (*Bv3_057690_fwmj*), and external alternative NAD(P)H-ubiquinone oxidoreductase B2 (*Bv8_181570_drqn*), while hypermethylated modification was found to increase the degradation rates of chloride channel protein CLC-b (*Bv5_113110_wkdm*), potassium transporter 8 (*Bv_005060_xrpy*), WAT1-related protein At3g28050 (*Bv6_149350_agyc*), histidine kinase 5 (*Bv2_045130_nadu*), transcription factor bHLH71 (Bv5_125640_rrpy), and calcium-dependent protein kinase 17 (*Bv5_108920_fgau*). These results suggest that m^6^A methylation negatively regulated the expression of some salt tolerance-related genes by influencing the transcript stability of these genes.

To validate the RNA-seq data, ten random genes were selected from DGEs with DMPs to detect their expression levels via qRT-PCR. The results show that eight of the ten genes were up-regulated and two were down-regulated, which is in good agreement with the RNA-seq data (Figure 7).

## 3. Discussion

Salt stress is one of the main environmental restrictions, and might cause osmotic stress, ion toxicity, or oxidative stress in plants [26,27]. Unlike other crops, sugar beet displays a relatively higher tolerance to saline alkali soil environments, which makes it easier for them to grow in such harsh environments. Plants undergo physiological and biochemical adjustments via transcriptional and post-transcriptional regulation to minimize the negative effects of salt stress. Based on the acquired results, we found that m^6^A modification was involved in the salt stress response in sugar beets by modulating the expression of genes associated with energy metabolism, transport, signal transduction, transcription factors, and cell wall organization (Figure 8). Collectively, we identified an important role of m^6^A in salt stress and suggest post-transcription regulation as an essential process for salt tolerance.

Energy is indispensable for plants as it enables them to absorb and transport salt ions, to synthesize osmotic regulators, to regulate transcripts, and to adjust the direction of metabolism in response to salt stress. Glycolysis is the anaerobic oxidation of hexose, which provides ATP and NADH to the organism, and is also the pathway by which pyruvic acid is prepared for aerobic oxidation [28]. Common hexoses like glucose and galactose exist in two predominant forms (α and β) in aqueous solution, but many enzymes of carbohydrate metabolism exhibit specificity for one anomer or the other [29]. Aldose 1-epimerase is a key enzyme that catalyzes the tautomerism of the α- and β-anomers of hexose. Subsequently, hexose is activated via phosphorylation into glycolysis, and hexokinase (HXK) catalyzes this reaction as a rate-limiting enzyme for glycolysis [30]. In the cytoplasm, fructose-bisphosphate aldolase (FBA) can form complexes with glyceraldehyde-3-phosphate dehydrogenase (GAPDH), and then, bind to mitochondria in response to the energy demands of cells in different states [31]. In addition, FBA can interact with subunit B of V-ATPase localized on the vacuole membrane to enhance the affinity of ATPase for ATP, providing more energy for the transmembrane transport of sodium ions in plants under salt stress [32,33]. The m^6^A-seq data revealed an increase in the expression of aldose 1-epimerase (*Bv5_119690_uayn*), *HXK1* (*Bv9_224670_aurc*), and *FBA5* (*Bv4_091040_cyuu*) under salt stress, while the extent of the m^6^A modification of their mRNAs decreased (Figure 7 and Appendix A). Moreover, the time course experiments confirmed that the lack of m^6^A enhanced the stability of *FBA5* mRNA (Figure 6). These results provide targets for m^6^A modification to participate in glycolysis regulation under salt stress.

Oxidative phosphorylation is the primary energy source of aerobic organisms and the main pathway by which cells produce ATP. The oxidation reactions of carbohydrates, lipids, and proteins are accompanied by the reduction of NAD^+^ and FAD to produce NADH and FADH_2_, which are then subjected to oxidative phosphorylation to transfer energy to ATP [34]. Electron transfer flavoprotein—ubiquinone oxidoreductase (ETFQO) is a component of the electron transport chain that, together with electron transfer flavoprotein (ETF), forms a short pathway that transfers electrons from mitochondrial flavoprotein dehydrogenases to the ubiquinone pool [35]. During stress situations, plant cells also use amino acids as alternative substrates of the electron donor protein isovaleryl CoA dehydrogenase (IVDH), and the released electrons are donated to the mitochondrial electron transport chain (mETC) [36]. In the present study, external alternative mitochondrial NAD(P)H: ubiquinone oxidoreductase B2 (*Bv8_181570_drqn*), *ETFQO* (*Bv3_057690_fwmj*), and *IVDH* (*Bv9_208200_roux*) were found to be up-regulated in m^6^A expression and down-regulated in m^6^A modification under salt stress (Appendix A). As expected, hypomethylation modification increased the retention rates of *Bv8_181570_drqn* and *Bv3_057690_fwmj*. Collectively, m^6^A modification may play an essential role in regulating sugar beet energy metabolism under salt stress.

Exposure to salt stress can be rapidly sensed by roots and using transmitted stress signals, and plants adapt their growth and development to environmental changes by reshaping their cellular transcriptional networks. In this study, the expression and methylation levels of many signal transduction-related genes were significantly altered under salt stress (Figure 3C and Figure 4E). As a phytohormone, cytokinin plays a crucial role in plant growth and development, as well as in response to biotic and abiotic stresses [37]. Cytokinin oxidase/dehydrogenase (CKX) can catalyze the irreversible degradation of cytokinin [38]. Cytokinins are sensed by membrane-localized histidine kinase receptors and activate transcription factors in the nucleus via phosphorylation [39]. M^6^A modification of the cytokinin dehydrogenase 7 (*Bv2_033900_crig*), histidine kinase 5 (*Bv2_045130_nadu*), transcription factor bHLH71 (*Bv5_125640_rrpy*), and basic leucine zipper 19 (*Bv4_094850_mmdf*) was up-regulated, while their expression was down-regulated in beets under salt stress. As expected, the mRNA stability of *Bv2_045130_nadu* and *Bv5_125640_rrpy* was also negatively regulated by m^6^A modification. In addition, KEGG analysis revealed that WRKY transcription factor 22 (*Bv5_099870_ypwu*) and MYB family transcription factor EFM (*Bv6_134560_dzpj*) were involved in the MAPK signaling pathway—plant and plant hormone signal transduction, respectively. The up-regulated expression of *Bv5_099870_ypwu* and *Bv6_134560_dzpj* in sugar beet was accompanied by their down-regulated methylation under salt stress. Phytohormones are widely involved in the regulation of the plant response to environmental stress. m^6^A modification negatively regulated phytohormone metabolism, signal transduction, and transcription factors at the post-transcriptional level, and affected the root response to salt stress in sugar beet.

Transport proteins, which include transporters, channels, and adenosine triphosphatase (ATPase) pumps, are a large class of membrane proteins that mediate chemical and signaling exchange within and outside biofilms [40]. Many of the DMPs in this study are located on transport-related genes (Figure 3C). The mRNA levels of potassium transporter 8 (*Bv_005060_xrpy*) and chloride channel protein CLC-b (*Bv5_113110_wkdm*) exhibited increased m^6^A methylation and decreased expression, and the stability experiments also confirmed that hypermethylation promoted the degradation of both genes. And the down-regulation of Bv_005060_xrpy was consistent with a reduction in potassium content (Figure 1G). In addition, the up-regulation of vacuolar amino acid transporter 1 (*Bv2_046760_tquf*), amino-acid permease BAT1 (*Bv6_143190_zduk*), nucleobase-ascorbate transporter 4 (*Bv9_222860_dmzk*), nitrate transporter NRT1/ PTR (*Bv5_098450_cqnk*), and organic cation/carnitine transporter 7 (*Bv4_085370_qpkj*) was negatively regulated by m^6^A modification under salt stress. Overexpression of the nucleobase-ascorbate transporter gene *MdNAT7* has been reported to enhance salt tolerance in apples [41]. These results suggest that m^6^A modification was involved in the salt stress response by modulating the transport and distribution of metal ions, nitrates, signaling molecules, amino acids, purines, and pyrimidines in sugar beet.

Another enrichment function of DMPs is cell wall-related genes. Cell walls are dynamic entities that can be remodeled during plant development and in response to abiotic and biotic stresses [42]. Cellulose is one of the main components of plant cell walls. Existing studies have shown that salt stress affects plant growth by inhibiting cellulose synthesis. Lignin is another major component of the cell wall, and abiotic stress can rapidly induce lignin biosynthesis and deposition on the surface of secondary cell wall cellulose polymers, which play an important role in the stabilization of the cell wall under salt stress [43]. Cellulose synthase A (CesA) complexes are responsible for the synthesis of cellulose, and cinnamoyl-CoA reductase (CCR) catalyzes the first specific committed step in lignin biosynthesis in plants [44]. At the same time, peroxidase is also an important lignin synthesis-related gene. In this study, we found that the expression of cinnamoyl-CoA reductase-like SNL6 (*Bv3_049630_kwyd*) was up-regulated, while the expression of cellulose synthase A catalytic subunit 3 (*Bv_006490_cpjn*) was down-regulated. The extent of their m^6^A modification also displayed a negative correlation with their expression. In addition, significant up-regulation of POD activity was observed in roots under salt stress (Figure 1A). These results suggest that m^6^A modification may modulate cell wall remodeling under salt stress by inhibiting cellulose synthesis and enhancing lignin biosynthesis in the roots of sugar beets.

Several studies on the role of the m^6^A modification of mRNA in salt tolerance have been published recently. In *Arabidopsis thaliana*, hypermethylation increases mRNA stability for transcripts encoding salt stress response proteins by inhibiting site-specific cleavage [45] and eliminating RNA secondary structures [19] in plant transcripts. Studies on cotton [46] and sweet sorghum [47] have shown similar results to Arabidopsis, with m^6^A positively modulating the expression of salt-tolerant genes. However, in addition to several salt-tolerant genes that were positively modulated by m^6^A modification, many more salt-tolerant genes in sugar beet exhibited negative regulation by m^6^A modification. It appears that hypomethylation also enhances mRNA stability under salt stress, which is in agreement with the results of poplar [48], although the exact mechanism remains unclear. Further, in *MTA* RNAi plants, m^6^A-containing genes showed higher translation efficiency than non-m^6^A-containing genes under chilling [49]. m^6^A modification in sugar beet may also directly regulate the translational efficiency of target genes. In the present study, more than half of the DMP-containing genes showed no differences in their expression patterns, while in contrast to our previous proteomic data, we found significant changes in protein abundance for some DMP-containing genes that were not differently expressed [50]. For example, hypermethylation did not affect the expression pattern of *Bv9_203470_nuyj*, a gene that encodes the pentatricopeptide repeat-containing protein At1g61870 (A0A0K9Q835) but significantly reduced the abundance of this protein. However, the clarification of this conjecture will require further study.

## 4. Materials and Methods

### 4.1. Cultivation and Treatment of Beet Seedlings

Seeds of the sugar beet cv. O68 were obtained from the Harbin Institute of Technology (Heilongjiang, China). The seeds were grown in plant hydroponic tanks with half-strength Hoagland solution under a 16 h/8 h light photoperiod at 24 °C (day) /18 °C (night) until the three-pairs-of-euphylla stage. The half-strength Hoagland solution was replaced with a fresh solution every three days. Following a previous study [51], the three-pairs-of-euphylla-stage seedlings were treated with 300 mmol·L^−1^ NaCl for 24 h, after which the roots were harvested and cleaned with a PBS buffer. Control seedlings were also prepared by treating the seedlings with just distilled water instead of NaCl. 

### 4.2. Measurement of Physiologic Indicators and Ion Content

The fresh root samples were directly used for physiological index detection. The content of Proline (#G0111W, Grace Biotechnology, Suzhou, China); the activity of POD (#G0107W, Grace Biotechnology, Suzhou, China), SOD (#G0104W, Grace Biotechnology, Suzhou, China), and CAT (#G0106W, Grace Biotechnology, Suzhou, China); and root activity (#G0124W, Grace Biotechnology, Suzhou, China) were detected according to standard protocols. The data were obtained using a BioTek Epoch (BioTek, Highland Park, IL, USA). The contents of sodium, potassium, and calcium were determined using an ICP Mass Spectrometer (Thermo Fisher, Bremen, Germany). All these experiments were conducted in triplicate for each treatment.

### 4.3. RNA Extraction, Library Construction, and Sequencing 

Fresh root samples were immediately frozen in liquid nitrogen for half an hour, and then, stored at −80 °C until further use. Total RNA was isolated and purified vis TRIzol^TM^ (#15596026, Invitrogen, Carlsbad, CA, USA) from frozen samples following the manufacturer’s procedure. The quantity and purity of total RNA were analyzed using NanoDrop ND-1000 (Agilent, Palo Alto, CA, USA) and Bioanalyzer 2100 (Agilent, Palo Alto, CA, USA) with RIN number > 7.0. Poly (A) RNA was purified from 100 μg total RNA using Dynabeads^TM^ Oligo (dT)_25_ (#61002, Thermo Fisher, Carlsbad, CA, USA), and then, fragmented into short pieces using a NEBNext^®^ Magnesium RNA Fragmentation Module (#E6150S, NEB, Ipswich, MA, USA) at 86 °C for 7 min. The cleaved RNA fragments were incubated at 4 ℃ for 2 h with an m^6^A-specific antibody (#202003, Synaptic Systems, Göttingen, Germany) in IP buffer (50 mM Tris-HCl, 750 mM NaCl, and 0.5% Igepal CA-630). The eluted m^6^A-containing fragments (IP) and non-immunoprecipitated fragments (input) were used to construct cDNA libraries for m^6^A-seq and RNA-seq, respectively. Finally, paired-end sequencing was performed with an average insert size of 300 bp (± 50 bp) using an Illumina Novaseq™ 6000 (LC-Bio Technology Co., Ltd., Hangzhou, China) following the vendor’s recommended protocol. The raw sequencing data were uploaded to the NCBI Short Read Archive (SRA) database (BioProject ID: PRJNA936097).

### 4.4. Analysis of Sequencing Data

Raw reads obtained from the sequencing of the IP and Input samples were processed to remove the contaminating primers/adaptors, low-quality bases, and undetermined bases using the fastp software (https://github.com/OpenGene/fastp, v0.19.4 accessed on 31 August 2018) with default parameters. The clean reads were then mapped to the reference genome (http://bvseq.boku.ac.at/Genome/index.shtml, RefBeet-1.2 accessed on 4 September 2021) via HISAT (http://daehwankimlab.github.io/hisat2, v2.0.4 accessed on 22 May 2016). The mapping reads were then utilized in exomePeak (https://bioconductor.org/packages/exomePeak, v2.16.0 accessed on 15 April 2019) to identify the m^6^A peaks in either the bed or bigwig format, which could then be visualized using the IGV software v2.13.2. MEME (http://meme-suite.org, v5.4.1 accessed on 10 September 2021) and HOMER (http://homer.ucsd.edu/homer/motif, v4.11 accessed on 24 October 2019) were used for de novo and known motif finding, followed by motif localization with respect to the peak. Peaks were annotated using the R package ChIPseeker (https://bioconductor.org/packages/ChIPseeker, v1.19.1 accessed on 12 December 2018) by interleaving them with the gene architecture. StringTie (https://ccb.jhu.edu/software/stringtie, v2.1.5 accessed on 30 September 2021) was used to calculate the FPKM (fragments per kilobase of transcript per million mapped reads) of all transcripts. The DEGs were selected with |log2 (fold change)| > 1 and *p* < 0.05 using edgeR (https://bioconductor.org/packages/edgeR, v3.24.3 accessed on 3 January 2019). All transcripts were annotated using the NCBI non-redundant protein database (Nr), and the GO and KEGG databases [52,53].

### 4.5. Transcript Stability Time Course

After the treatment with 300 mmol L^−1^ NaCl (st) or distilled water (ck) reached 24 h, half of the seedlings (st + ck) were directly harvested, and the other half (st + ck) were subjected to further treatment with 0.6 mM cordycepin plus 10 μM actinomycin D for another 24 h before harvesting. RNA was then extracted from the four groups of plants. Subsequently, total RNA was used to synthesize the cDNA using High-Capacity cDNA Reverse Transcription Kits (#4368813, Thermo Fisher, Foster City, CA, USA). The qPCR reactions were performed using iTaq Universal SYBR^®^ Green Supermix (#1725121, BIO-RAD, Hercules, CA, USA) via a CFX Real-time PCR system (BIO-RAD, Singapore, SG). The primer sequences are listed in Appendix A. According to our previous study (Li et al., 2020b), *PP2A* plus *UBQ5* were used as endogenous controls. The relative expression quantity was calculated using the comparative 2^−ΔΔCt^ method [50].

### 4.6. Statistical Analysis

All experiments were conducted in triplicate (*n* = 3), and the statistical analysis of the data from ck and st was performed by analyzing the independent-samples via *t*-test using SPSS (version 22.0). Statistically significant differences were considered at the *p* < 0.05 level. Data are presented as the mean ± standard deviation (SD) of the three replicates.

## 5. Conclusions

In summary, the results from m^6^A-seq reveal that salt stress induces changes in RNA methylation, affecting the expression of many genes in the roots of sugar beet. The combined analysis of m^6^A modifications and the expression patterns of these genes indicated that most salt-tolerant genes were negatively regulated by m^6^A methylation. It is interesting to note that m^6^A methylation has been implicated in the regulation of key genes involved in energy metabolism, transport, signal transduction, transcription factors, and cell wall organization. Furthermore, BvALKBH1A and BvALKBH10A may be located upstream of this regulatory network, but elucidating this conjecture will require further investigation. These findings have the potential to provide a better understanding of the epigenetic mechanisms responsible for salt tolerance in sugar beet, as well as to uncover gene candidates responsible for improved stress resistance in sugar beet planted in high-salinity soils.

## Figures and Tables

**Figure 1 ijms-24-11659-f001:**
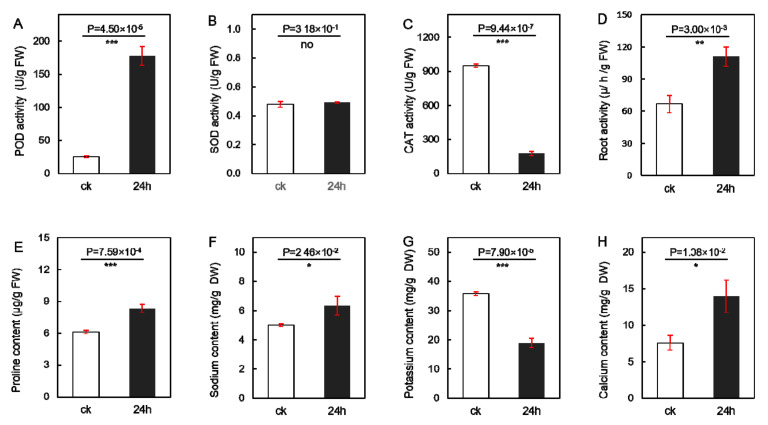
Changes in physiological and ionic responses in sugar beet to salt stress. (**A**) Peroxidase activity. (**B**) Superoxide dismutase activity. (**C**) Catalase activity. (**D**) Root activity. (**E**) Proline content. (**F**) Sodium content. (**G**) Potassium content. (**H**) Calcium content. Data are presented as means ± SDs of three biological repetitions. ‘*’, ‘**’, and ‘***’ indicate significant differences at *p* < 0.05, *p* < 0.01, and *p* < 0.001, respectively, as determined by Student’s *t*-test using SPSS software (version 22.0).

**Figure 2 ijms-24-11659-f002:**
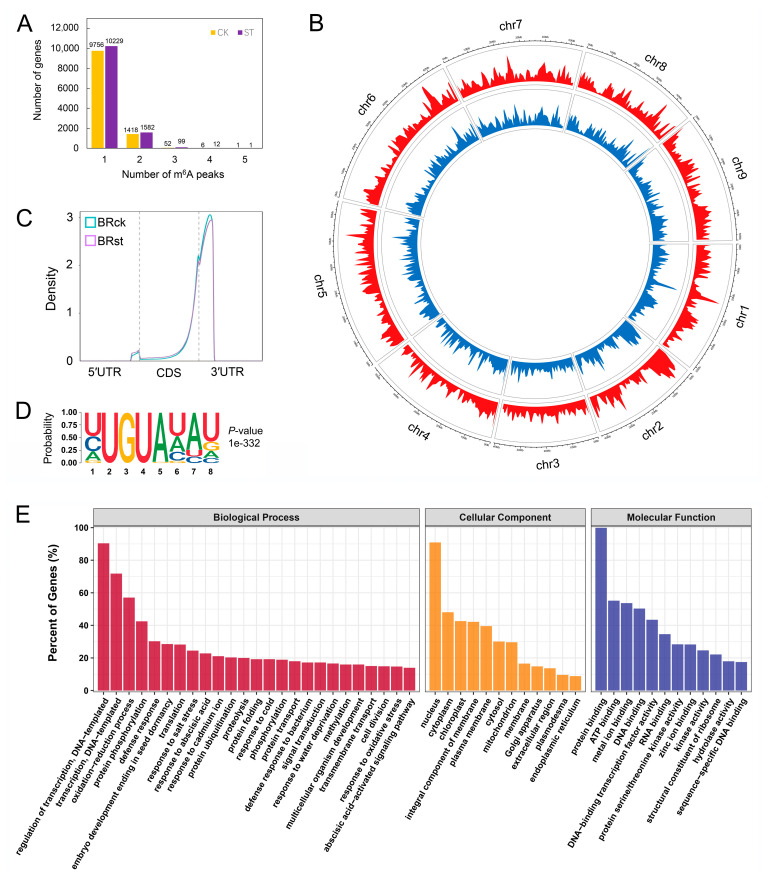
Characteristics and extent of m^6^A modification in sugar beet roots. (**A**) Distribution of the number of m^6^A modifications in genes. (**B**) Circos plot of the distance and expression abundance of m^6^A peak in sugar beet chromosomes. The tracks are “peaks in salt-treated (st) plants (blue)”, “peaks in control (ck) plants (red)”, and “distance and size in the known chr1–9 chromosomes” from inside outwards. Unplaced genes are not shown in the graph; peak height represents the multiplication of the IP sample by the input sample. (**C**) Distribution and density of m^6^A peak in the transcript structure consisting of 5′ UTR, CDS, and 3′ UTR. (**D**) Annotation of identified m^6^A high-confidence peaks in URUAY motif in samples. (**E**) Gene ontology (GO) annotation for peaks in st and ck plants.

**Figure 3 ijms-24-11659-f003:**
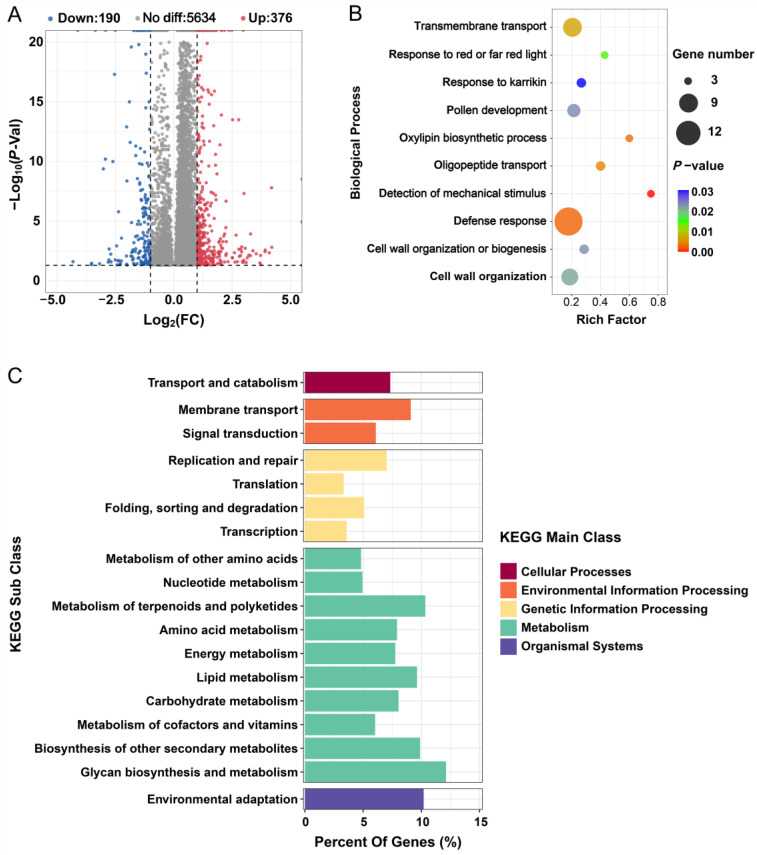
Analysis of differentially modified m^6^A methylation peaks (DMPs) in sugar beet comparing salt treatment (st) and control (ck) conditions. (**A**) Volcano plot of up- (red) and down- (blue) regulated peaks in st and ck in sugar beet roots. The dotted line represents the DMP threshold |Log2(fold change)| ≥ 1 and *p* < 0.05. (**B**) Top 10 gene ontology terms for enrichment of DMPs in biological process. (**C**) KEGG analysis of DMPs comparing salt treatment and control conditions.

**Figure 4 ijms-24-11659-f004:**
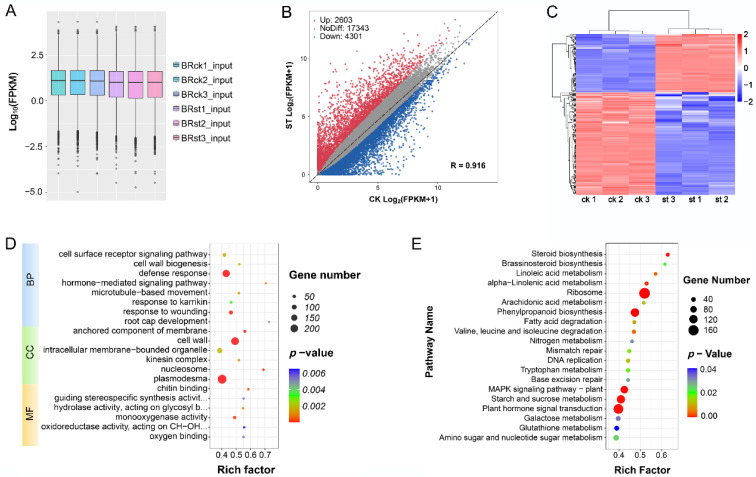
Identification and analysis of differentially expressed genes in sugar beet under NaCl stress. (**A**) Box plot showing the gene expression patterns for each input sample. (**B**) Scatter plots of up- (red) and down- (blue) regulated genes in st and ck in roots. (**C**) Heatmap displaying the correlation of gene expression in three biological replicates under st and ck conditions. (**D**) The top 20 significantly enriched GO terms for DEGs under salt stress. BP: biological process, CC: cellular component, MF: molecular function. (**E**) KEGG analysis for DEGs comparing salt treatment and control conditions.

**Figure 5 ijms-24-11659-f005:**
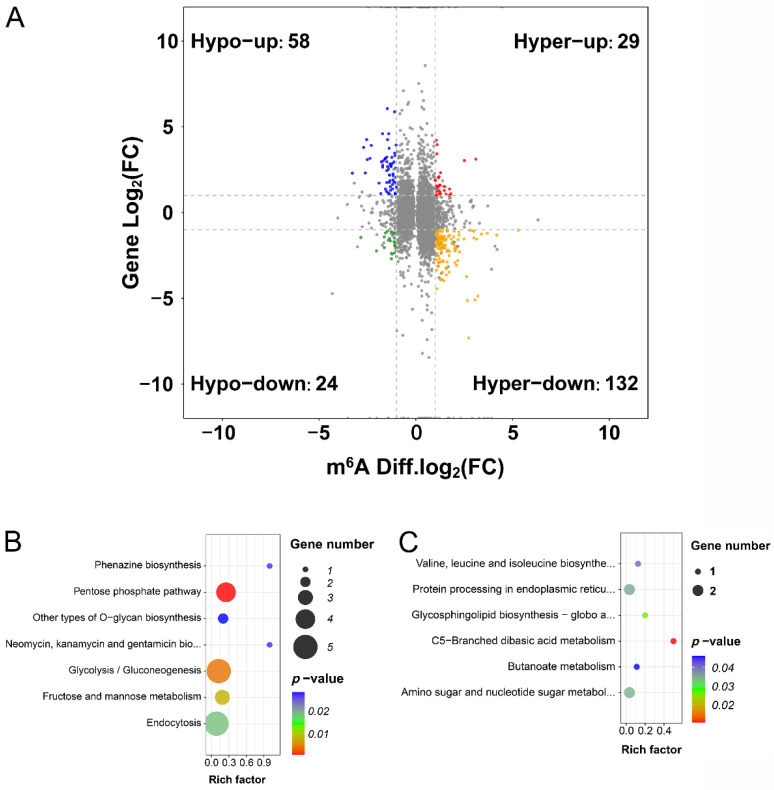
Integrating analysis of differentially methylated peaks and differentially expressed genes in sugar beet after NaCl treatment. (**A**) Four-quadrant graph representing the relationship between m^6^A methylation and gene expression. Red dots: hypermethylation and up-regulated genes; Yellow dots: hypermethylation and down-regulated genes; Blue dots: hypomethylation and up-regulated genes; Green dots, hypomethylation and down-regulated genes. (**B**) KEGG analysis of genes whose expression levels were negatively correlated with m^6^A modification levels. (**C**) KEGG analysis of genes whose expression levels were positively correlated with m^6^A modification levels.

**Figure 6 ijms-24-11659-f006:**
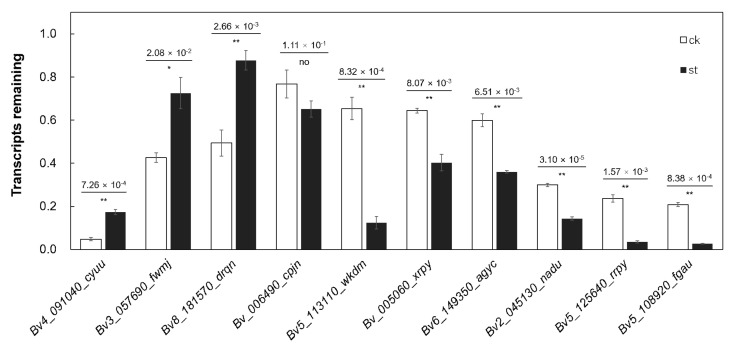
Effect of m6A modification abundance on transcript stability. Data are means ± SDs of three biological repetitions. ‘*’ and ‘**’ indicate significant differences at *p* < 0.05 and *p* < 0.01, respectively, as determined by Student’s *t*-test using SPSS software. The white bars represent the control group and the black bars represent the salt-treated group.

**Figure 7 ijms-24-11659-f007:**
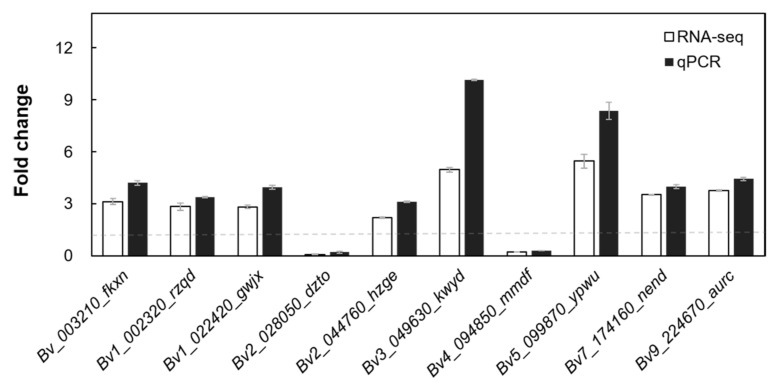
Comparison of qPCR and RNA-seq results. Data are means ± SDs of three biological repetitions. The white bars represent RNA-seq and the black bars represent qPCR. The dashed gray line represents FC = 1.

**Figure 8 ijms-24-11659-f008:**
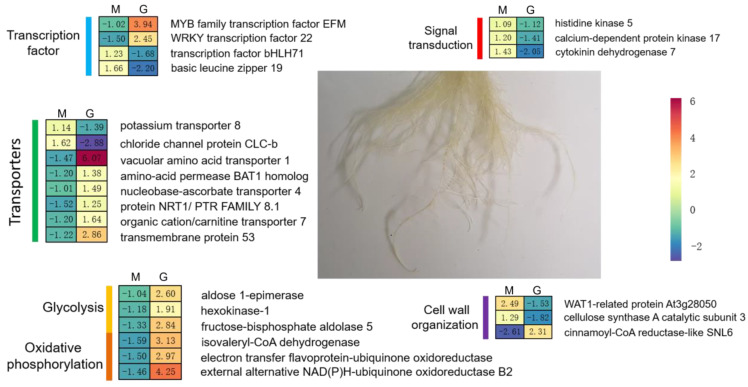
A working model of regulatory networks proposed by m^6^A modification in beet roots under salt stress. ‘M’ represents the enrichment of the m6A peak, ‘G’ represent the expression of the gene. The heatmap displays Log_2_ (fold change between st and ck) for ‘M’ and ‘G’.

**Table 1 ijms-24-11659-t001:** RNA methylation-related genes in sugar beet.

Gene_Id	Type	Homologous Gene	log_2_ (FC)	*p*-Value	Up/Down
*Bv5_103730_enox*	writer	*AtMTA*	−0.33	0.17	no dif.
*Bv3_054970_hfjn*	writer	*AtMTB*	−0.39	0.00	no dif.
*Bv5_117690_arcy*	writer	*AtFKBP12*	0.15	0.00	no dif.
*Bv5_110090_noir*	writer	*AtVIR*	−0.86	0.00	no dif.
*Bv5_121260_sckp*	writer	*AtHAKAI*	−0.30	0.44	no dif.
*Bv6_150770_huzh*	eraser	*AtALKBH1A*	−1.01	0.00	down
*Bv7_157650_ryeg*	eraser	*AtALKBH1D*	0.24	0.00	no dif.
*Bv7_169620_pkhc*	eraser	*AtALKBH1D*	−0.45	0.06	no dif.
*Bv8_184320_kacr*	eraser	*AtALKBH2*	0.08	0.05	no dif.
*Bv5_102160_pgse*	eraser	*AtALKBH8A*	−0.63	0.01	no dif.
*Bv3_051230_eskg*	eraser	*AtALKBH9A*	0.71	0.00	no dif.
*Bv4_083160_sqec*	eraser	*AtALKBH6*	0.08	0.00	no dif.
*Bv6_130050_njrf*	eraser	*AtALKBH5*	−0.27	0.78	no dif.
*Bv7_164580_swwm*	eraser	*AtALKBH8B*	−0.35	0.35	no dif.
*Bv7_179400_uxaj*	eraser	*AtALKBH10A*	−2.93	0.00	down
*Bv2_036020_nmug*	reader	*AtECT10*	0.22	0.00	no dif.
*Bv3_056220_tirq*	reader	*AtECT3*	−0.37	0.00	no dif.
*Bv3_059680_euso*	reader	*AtECT6*	0.12	0.00	no dif.
*Bv5_101530_jzsk*	reader	*AtECT12*	−0.27	0.82	no dif.
*Bv8_181150_pemc*	reader	*AtECT11*	−0.33	0.01	no dif.
*Bv8_187630_mced*	reader	*AtCPSF30*	−0.22	0.65	no dif.

## Data Availability

All sequencing data were deposited in the NCBI Short Read Archive (SRA) database under the BioProject ID: PRJNA936097. Relevant supporting data can be found within the article and Appendix A.

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
