# Peer review of "Unique Features of the m6A Methylome and Its Response to Salt Stress in the Roots of Sugar Beet (Beta vulgaris)"

_ijms, 2023, doi:10.3390/ijms241411659_

Round 1

Reviewer 1 Report

The result section is missing the validation step. Validating the reliability of RNA-seq data is essential to ensure the accuracy of the results. This is typically done through independent experimental methods such as quantitative PCR (qPCR) or other targeted gene expression assays. 

Author Response

Thank you for your comments. We supplemented the qPCR experiments as requested and added them to the section 2.6.

Reviewer 2 Report

It is a very well written paper and the results are very interesting. I have no concern for not accepting the paper as it is.

I just found that at line 86 the name of the gene should be in Italicum

Author Response

Thank you for your advice. We have corrected it as requested.

Reviewer 3 Report

The authors of the manuscript titled “Unique features of the m6A methylome and its response to salt stress in the roots of sugar beet (Beta vulgaris)” The current experiment aimed to investigate m6A-seq was used to investigate changes in the RNA methylation profile of roots under salt stress in sugar beet.

General comments

Overall, the study is well-designed and presented in a good way.

Keywords

The authors are requested to correct the first keyword.

Title

The title of the study is okay.

Abstract

The abstract of the study needs further improvement.

The authors requested to add the methodology in the abstract section.

The authors failed to add the quantitative data of the studied parameters and even they did mention the studied parameters at all in the abstract section of the manuscript. Therefore, the authors requested to add the objective of the current study to make the abstract section easy to understand.   

Introduction

The authors are requested to rewrite the objectives of the study and make it easy for the readers.   

Results

The authors are requested to add quantitative data on the studied parameters of section 2.1.

Discussion

The authors are requested to add quantitative data of the studied parameters.

Conclusion

The authors are requested to add a Conclusion section to make this study-friendly reading easy to understand.

Author Response

Keywords

The authors are requested to correct the first keyword.

Response: The first keyword has been corrected to “m6A-sequencing”

Abstract

The abstract of the study needs further improvement.

The authors requested to add the methodology in the abstract section.

The authors failed to add the quantitative data of the studied parameters and even they did mention the studied parameters at all in the abstract section of the manuscript. Therefore, the authors requested to add the objective of the current study to make the abstract section easy to understand.   

Response: Thank you for your advice. The abstract section has been improved as required.

Salt is one of the most important environmental factor in crop growth and development. N6-methyladenosine (m6A) is an epigenetic modification that regulates plant-environment interaction at transcriptional and translational levels. Sugar beet is a salt-tolerant sugar-yielding crop, but how the m6A modification affects the response to salt stress remains unknown. In this study, m6A-seq was used to explored the role of m6A modification in response to salt stress in sugar beet (Beta vulgaris). Transcriptome-wide m6A methylation profiles and physiological responses to high salinity have been investigated in beet roots. After treatment with 300 mM NaCl, the activities of peroxidase and catalase, the root activity, and the contents of Na+, K+ and Ca2+ in the roots were significantly affected by salt stresses. Compared with control plants, 6,904 differentially expressed genes (DEGs) and 566 differentially methylated peaks (DMPs) were identified. Association analysis revealed that 243 DEGs contained DMP, and 80% of these DEGs had expression patterns that were negatively correlated with the extent of m6A modification. Further analysis verified that m6A methylation may regulate the expression of some genes by controlling their mRNA stability. Functional analysis revealed that m6A modifications primarily affect the expression of genes involved in energy metabolism, transport, signal transduction, transcription factors, and cell wall organization. This study provides evidence that a post-transcriptional regulatory mechanism mediates gene expression during salt stress by affecting the stability of mRNA in the root.

Introduction

The authors are requested to rewrite the objectives of the study and make it easy for the readers.  

Response: The objectives of the study have been rewritten as required.

Sugar beet have been reported to have the ability to store salt absorbed from the environment in their cells to improve soil salinization [8]. The roots are directly exposed to salt and play an important role in adaptation to salt stress. However, the dynamic response of m6A modifications to transcripts under salt stress in sugar beet roots is still unknown. In this study, m6A-seq was used to investigate changes in the RNA methylation profile of roots under salt stress in sugar beet.

Results

The authors are requested to add quantitative data on the studied parameters of section 2.1.

Response: ​Table S1 was added to give original quantitative data on physiological indicators and ionic content under salt stress. In addition, ​We have added fold change data to the manuscript section 2.1. “As for the activity of protective enzymes, plants treated with salt showed a seven-fold increase in peroxidase activity (Fig. 1A and Table S1), while displaying no significant changes in superoxide dismutase activity (Fig. 1B), and a five-fold decrease in catalase activity.”

Discussion

The authors are requested to add quantitative data of the studied parameters.

Response: ​Table S1 was added to give original quantitative data on physiological indicators and ionic content under salt stress.

Conclusion

The authors are requested to add a Conclusion section to make this study-friendly reading easy to understand.

Response: The conclusion section was added as required.

  1. Conclusions

In summary, the results from m6A-seq reveal that salt stress induces changes in RNA methylation, affecting the expression of many genes in the roots of sugar beet. The combined analysis of m6A modifications and expression patterns of these genes indicated that most salt-tolerant genes were negatively regulated by m6A methylation. It is interesting to note that m6A methylation has been implicated in the regulation of key genes involved in energy metabolism, transport, signal transduction, transcription factors, and cell wall organization. Furthermore, BvALKBH1A and BvALKBH10A may be upstream of this regulatory network, but elucidating this conjecture will require further investigation. These findings have the potential to provide a better understanding of the epigenetic mechanisms responsible for salt tolerance in sugar beet, as well as uncover gene candidates responsible for improved stress resistance of sugar beet planted in high-salinity soils.

Round 2

Reviewer 1 Report

The missing data was provided and the manuscript was improved and qualified for publication.

Reviewer 3 Report

I accept the manuscript in its present form.